# Titanite-Containing Mineral Compositions and Their Chemical Treatment with Preparation of Functional Materials

**DOI:** 10.3390/ma13071599

**Published:** 2020-04-01

**Authors:** Lidia G. Gerasimova, Anatoly I. Nikolaev, Ekaterina S. Shchukina, Marina V. Maslova

**Affiliations:** 1Tananaev Institute of Chemistry of Kola Science Centre, Russian Academy of Sciences, 26aFersman Street, 184209 Apatity, Russia; l.gerasimova@ksc.ru (L.G.G.); m.maslova@ksc.ru (M.V.M.); 2Nanomaterials Research Centre of Kola Science Centre, Russian Academy of Sciences, 14 Fersman Street, 184209 Apatity, Russia; a.nikolaev@ksc.ru

**Keywords:** mineral compositions, chemical treatment, titanite, concentrate, decomposition, leaching, synthesis, rutile, anatase, silica

## Abstract

The waste of apatite-nepheline ore processing was chosen as the material of study for the present investigation. The chemical and phase compositions have been analyzed and the route of the new technology has been developed. Treatment of the waste with diluted hydrochloric acid enables to separate apatite, nepheline, titano-magnetite minerals from titanite (CaSiTiO_5_). The obtained titanite concentrate contains 30–32% of titanium dioxide. Interaction of titanite with hydrochloric acid under heating and stirring conditions results in calcium leaching. The titanite decomposition is accompanied by titanium and silica oxides precipitation. The resulting solid has been used as a precursor for the synthesis of functional materials. Mechanochemical activation of the precursor provides the structural and morphological disorder of the initial particles. Thermodynamic stability of activated particles is achieved by chemisorption or roasting.

## 1. Introduction

The utilization of the industrial wastes from the processing of multicomponent mineral ores is a more complicated and expensive task than storing them in the dumps. At the same time, maintenance of environmental standards for the functioning of enterprises is placed in the second place. In the Murmansk region, where the mining industry is developing rapidly, the amount of waste increases by millions of tons annually. The Khibiny deposits of complex apatite-nepheline ores (ANO) are the largest ones not only in Russia, but also in the whole world. Five main rock-forming minerals are specified in the ore: apatite, nepheline, sphene, aegirine, and titano-magnetite (Table 1), in which the content varies within certain ranges depending on the ore-bearing rocks [1]. However, the modern ANO processing aims at deeper recovery of apatite and partially nepheline. The rest minerals are almost not recovered because of the absence of demand for them; and therefore, they are disposed at tailings storage facilities [2].

Among the disposed minerals, calcium titanite-titanosilicate (CaTiSiO5) containing titanium represents a practical interest. Titanium metal and its compounds are diversely used in the industries producing products of various purpose.

Scientific and practical interest is associated with studies on titanite processing with preparation of various materials: non-toxic hardening agents [3,4], mineral-like alkaline titanosilicates with skeleton structure [5,6,7,8,9,10,11,12] having sorption properties, functional fillers of plastic materials, rubber [13], sealant [14]. Mineral and synthetic waste products are widely used in the building industry as concrete aggregates [15], paints on the organic and water bases [16], special additives, in particular for imparting fire-retardant and photocatalytic properties for products. The use of industrial waste as fillers is covered in great detail in the monograph [17].

### Titanite in Apatite-Nepheline Ore and its Position during the Ore Processing

Titanite is a typical apatite-nepheline mineral associated with minor rock-forming types. Titanite inclusions occur in apatite-nepheline (a) and apatite-titanite (b) ores of the Koashva deposit [18] (Figure 1).

Recovery of titanite mineral concentrate and its chemical treatment is described in numerous published papers in monographs [2,19,20]. In apatite-nepheline ore mass, titanite mineral content varies within 2.5–3 wt %. According to the conventional technology [2,20], titanite is recovered as a concentrate from nepheline flotation tails representing a mineral composition with average contents, wt%: 41—titanite, 27—nepheline, 20—apatite, 12—aegirine. Complexity of physical and chemical separation of the minerals is driven by large process streams and low titanite recovery—10–15% [21,22,23]. Sometimes, ore layers contain areas with titanite accumulations in the form of plate and wedge-shaped crystals characterized by crystallographic individuality (Figure 2). Based on the published data [24], the main peaks on X-ray patterns of titanite samples (Å) are the following: 3.20; 2.98; 2.59. The crystals have adamantine luster. Hardness by Moose—5–5.5. Density—3.4–3.54 g/sm^3^ [25].

This article describes results of the studies on development of a new option for processing of a titanium-containing mineral composition (TMC) with recovery of titanite concentrate, which is further used for the preparation of a weather-resistant mineral pigment and a synthetic titanium-containing precursor for synthesis of hybrid fillers for high-temperature glues and joint sealants.

## 2. Materials and Methods 

### 2.1. Materials

The study subject matter is a titanite-bearing mineral composition in the form of a sand fraction (sand grain size is 0.5–1.5 mm) produced after nepheline flotation. Mineralogy of the subject matter is titanite (10–50 wt %), nepheline (15–50 wt %), fluorapatite (5–20 wt %), aegirine-augite and potassic magnesio-arfvedsonite (5–30 wt %), orthoclase and titano-magnetite (up to 5 wt %). The average chemical composition of titanite is (wt %): TiO_2_ 39 ± 3, SiO_2_ 30.1 ± 0.7, CaO 27 ± 2, FeO 1.1 ± 0.4, Nb_2_O_5_ 1 ± 1, Na_2_O 0.7 ± 0.6, SrO 0.4 ± 0.2, Ce_2_O_3_ 0.4 ± 0.2, Al_2_O_3_ 0.3 ± 0.2, La_2_O_3_ 0.1 ± 0.1.

The ore blocks were crushed in a jaw crusher and milled in a 1.5 kW AGO-2 planetary mill (NOVIC, Novosibirsk, Russia) until the powder fraction was less than 40 μm. The chemical composition of the milled sample was determined using a FT IR 200 spectrophotometer (Perkin Elmer, Waltham, MA, USA).

### 2.2. Methods

The TMC processing operations studied by the authors is shown in Figure 3.

For crystal grain decomposition, TMC was ground in a ball mill at grinding media to concentrate ratio of 5:1 during 1 h [26]. After grinding, TMC was treated to remove acid-soluble minerals (apatite and nepheline) with hydrochloric acid (HCl—37.5%) diluted to 30 and 50 g/L (Table 2). While mixing the acid, a TMC charge was added to achieve a mass ratio S:V_liq_ = 1:3–4. The suspension was mixed in a magnetic stirrer at a temperature of 20 °C and 40 °C for 3 h. Then the precipitate was separated by filtering, and washed from acidic mother liquor with distilled water (Tananaev Institute of Chemistry, Apatity, Russia). Moisture was removed from the precipitate by drying at 105 °C. The recovered concentrate contains 80% of titanite, which corresponds to titanium content of 31.5% in terms of TiO_2_. The multicomponent filtrate was disposed of to obtain an amorphous precipitate, which was calcined at 500 °C. According to powder X-ray diffraction analysis, the obtained composition consists of aluminum and calcium phosphates, as well as silica (Figure 4). The use of the composition in mixture of concrete increases its strength characteristics, which is confirmed by the authors of the publication [27].

Acid decomposition of titanite concentrate was also conducted with 32% hydrochloric acid. A concentrate charge was gradually added to the acid heated up to 80 °C until titanite mass to acid volume ratio reached S:V_liq_ = 1:3. Then the suspension was heated until boiling, and mixed for 5 h while returning the produced steam-gas mixture (SGM) to the reaction area. Further holding for the next 4 h involved stripping of SGM and its condensation with hydrochloric acid production. The acid was used in the process cycle. Under the conditions described above, calcium, titanium, and silicon leach into a liquid phase; and depending on their solubility in a multi-component system TiO_2_-SiO_2_-CaO-HCl, solid and liquid phases develop. The X-ray fluorescence analysis shows that the solid phase consists of amorphous silica and titanium dioxide with anatase-rutile structure—DTK (Figure 5). Calcium concentrates in the liquid phase, where it is recovered by crystallization in the form of calcium chloride [28]—CaCl_2_∙nHCl∙mH_2_O (n: 0.1–0.15; m: 1–1.5).

A weather-resistant mineral pigment and a filler were produced by ultra grinding (mechanic activation) of titanite concentrate and DTK correspondingly in the Pulverisette 7 planetary ball mill (Fritsch Germany). The mill has two grinding bowls (50 mL each) with balls of diameter 10 mm. Material of bowl walls and balls—titanium. Bowl rotation speed—750 rpm. Balls to ground material ratio = 10:1. Grinding time—1 h [29,30].

Phase composition of the recovered solid phases is determined with the Shimadzu XRD-6000 X-ray diffractometer. Particle morphology—with the Philips XL 30 scanning electron microscope. Surface properties of the produced samples were determined with the TriStar 3020 device using the BEI and BJH methods based on nitrogen adsorption/desorption. Composition of the solid phases was studied by way of X-ray fluorescence analysis with the MAKS-GV spectroscope.

## 3. Results

Chemical treatment of TMC with diluted HCl solution leads to dissolution of apatite and nepheline based on the following reactions:apatite—Ca_5_(PO_4_)_3_F + 10HCl = 5CaCl_2_ + 3H_3_PO_4_ + HFnepheline—(NaK)_2_Al_2_O_3_∙2SiO_2_ + 8HCl = 2NaCl + 2KCl + 2AlCl_3_ + 2SiO_2_∙nH_2_O

The studies showed feasibility of the process with hydrochloric acid concentrated to 50 g/L HCl and temperature 40 °C (Table 2). Titanite and aegirine do not dissolve under these conditions.

### 3.1. Preparation of a Weather-Resistant Pigment from Titanite Concentrate

The pigment preparation method is based on a concentrate ultra grinding process, which is accompanied not only by changes in particle size distribution, but also by transformation of particle surface due to mechanical activation. The activation stage represents a certain interest illustrated by an opportunity for preparation of titanium dioxide alloy powders of various purpose: for photocatalysis, anti-bacteriological materials, composites [31]. In the case under consideration, a high-energy impact is accompanied by disturbance of structural order in titanite grains and crystals, which leads to formation of a shock-resistant chemically active surface layer of microparticles. Mechanical activation of the powder can be visually observed and confirmed by a whiteness indicator of original (60%) and ground (87%) material pre-set with the X-RtteSP-62 spectrophotometer. Figure 6 shows data of the energy-dispersive analysis of concentrate particles ground in the planetary mill. In this case, the intensity of Si response is over 9000, Ca—7000, and Ti—6000 (Figure 6a), while these indicators for the original sample are 110, 170, and 100 correspondingly (Figure 6b). 

Images of titanite concentrate particles before and after grinding in the planetary mill are shown in Figure 7. 

High degree of the concentrate particles amorphization is manifested in the indicators characterizing its particle surface and porous system. Specific surface area of mechanically activated titanite becomes almost five times larger. Pore volume also increases mainly because of increase in the amount of mesopores (Table 3).

Because of a high degree of the upper layer loosening caused by ionization and amorphization, the materials ability to adsorb modifying substances grows. The process of surface modification is widely used in preparation of pigments and fillers of various purpose. Moreover, the specialized publications [32] note that this process is one of the most important in the pigment production technology. By selecting modifiers and conditions for their application, the properties of the final products can be varied over a wide range and their fields of application can be expanded accordingly. Sorption mechanism has physical and chemical nature, i.e., filling of structural defects with a substance and surface complexation that lead to the transformation of a surface layer, and its modification.

Phosphorus acid was chosen for experiments. This acid reacts with the components of shock-resistant titanite surface with the formation of hard soluble calcium and titanium phosphates improving whiteness of the final product, and its resistance to moisture and temperature variations. Consumption of H_3_PO_4_ corresponds to 0.5% of monohydrate in relation to titanite mass. The mixture was dispersed in the KM-1 vibrating mill during 1.5 h. Thermal treatment was conducted at 200 °C for fixation of surface new-growths on titanite particles. Whiteness of the modified pigment increased from 87% to 92.5%. Refraction index—2.0 c.u., which is comparable to zinc white (ZnO).

The developed titanite technology with preparation of a mineral pigment [33] is patented and implemented at pilot scale with production of the WD-VA water-dispersion vinylacetate paint, WD-AC water-dispersion acrylic paint of various colors. Such paints cost 15–20% cheaper than similar products produced from titanium dioxide. 

### 3.2. Preparation of Synthetic Titanium-Bearing Filler for High-Temperature Glues and Joint Sealants

Formulations of many sealants include various additives having a certain function each. Collectively, such additives increase the sealant dielectric and thermostabilizing properties, its resistance to radiation and chemicals, as well as resistance to bending, which is critically important for reliable internal and external sealing of equipment operating under extreme conditions [34,35,36]. The most popular additives are titanium dioxide and silica [37]. They are usually added as individual additives [17,38]. The compositions developed during their collective settlement represent a certain interest. Effect from valuable factors of such compositions dramatically exceeds a simple sum of effects from each component because of synergy.

When titanite concentrate reacts with concentrated hydrochloric acid, a composite precipitate develops based on the formula: CaSiTiO_5_ + 2HCl = [(TiO_2_ + SiO_2_)∙nH_2_O]sol. + [CaCl_2_]liq. (n = 0.4–0.6). This precipitate is the subject matter of our studies. Precipitate composition—DTK, wt %: TiO_2_—54.19; SiO_2_—44.11; Al_2_O_3_—0.47; P_2_O_5_—0.23. Weight loss at 350 °C—5.05 wt %. In this case, phase composition remains the same. The calcined material was mechanically activated for disaggregation and preparation of nanosized particles. Besides, its is known that a high-energy impact on particles from heavy balls having impact and shear action cause a high degree of structural and surface disorder because of the formation of an intermediate metastable phase TiO_2_II with its further transformation to rutile as follows: anatase—TiO_2_II—rutile [39,40].

Thermodynamic instability of the system was localized by thermal treatment at 800 °C for 2 h. After baking, the product contains rutile and silica phase—TSK (Figure 8).

Over 90% of the TSK powder is represented by a fairly narrow size fraction 0.4–1 µm (Figure 9). Specific surface area of the composition (S_spec._) corresponds to 10.05 m^2^/g. Apparently, the main contribution to this indicator is made by silica amorphous phase (Table 4).

TSK particles morphology (S_spec._, V_pores_) along with their technical characteristics (oil adsorption power—M, and refraction index) close to titanium dioxide (Technical terms TT 2321-001-04694196-2016 Titanium dioxide brand TRG) enables to conclude that TSK can be used as an efficient filler in production of opaque adhesives and sealants with high adhesion degree for gluing and sealing metal surfaces.

## 4. Discussion

The titanite-containing mineral compositions (TMC), which are disposed as apatite-nepheline ores processing tails, can be chemically treated with recovery of titanite concentrate for the synthesis of titanium functional materials. The TMC chemical treatment enables to increase titanium yield to final product (concentrate) almost by five times. By mechanical activation of the concentrate, a finely dispersed mineral pigment has been produced. The pigment was modified with titanium and calcium phosphates for improving its whiteness, water-resistance, and weathering stability. The method of titanite concentrate decomposition with hydrochloric acid has been developed. This method is based on a waste-free technology involving calcium leaching into the solution, crystallization of calcium chloride, and recovery of hydrochloric acid for reusing by evaporation. The precipitated solid, composed of titanium and silica oxides, was used as titanosilicate precursor for preparation of a filler for glues and sealants of specialized application. The investigation showed that grinding by a high-energy impact and shear action on the precursor causes structural and morphological damage to the particles, resulting in an intermediate metastable phase TiO_2_II formation. Because of this, the transformation “anatase → TiO_2_II → rutile” is achieved at a lower temperature i.e., 150–200 °C, than in the case without grinding.

## 5. Conclusions


Almost all apatite deposits of the Khibiny massifs contain sufficient reserves of (apatite)-nepheline-titanite ore, which forms lens-like bodies up to 50-m thick and above 5 km long. This is a good titanium source that can be tapped without the traditional flotation schema, but by using only acidic cleaning from soluble impurities of apatite and nepheline;A new technology of hydrochloric acidic processing of titanite is now developed. It allows extraction of about 90 wt % of Ti and Si into hydrated titanosilicate precipitate TDS, while Ca remain in the chloride solution;The calcium passing into the solution after acid decomposition of the titanite-containing concentrate was evaporated to obtain crystalline calcium chloride, which can be used as a defroster. The hydrochloric acid is collected in the acid decomposition process, and can be used in circulation, thus the scheme becomes practically waste free;The obtained hydrated titanosilicate precipitate, composed of titanium and silica oxides, was used as a precursor for the preparation of a filler for glues and sealants of specialized application;The finely dispersed mineral pigment has been produced by the mechanical activation of the titanite-containing concentrate. The pigment was modified with titanium and calcium phosphates for improving its whiteness, water-resistance, and weathering stability.


## Figures and Tables

**Figure 1 materials-13-01599-f001:**
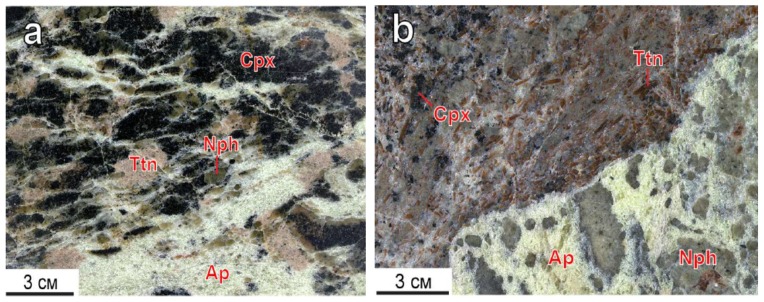
Titanite segregations in apatite-nepheline (**a**), apatite-titanite (**b**) ores of the Koashva apatite deposit, the Khibiny massif. Ap—fluorapatite, Cpx—clinopyroxene, Nph—nepheline, Ttn—titanite.

**Figure 2 materials-13-01599-f002:**
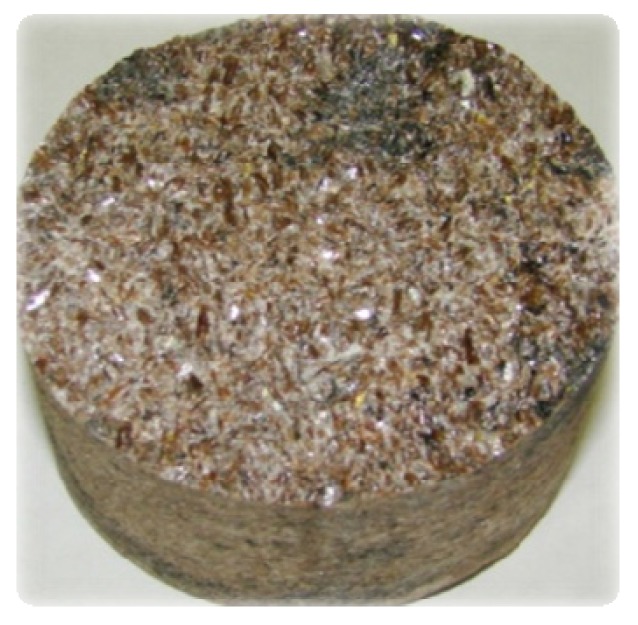
Titanite crystals in a rock sample (crystal color—light brown).

**Figure 3 materials-13-01599-f003:**
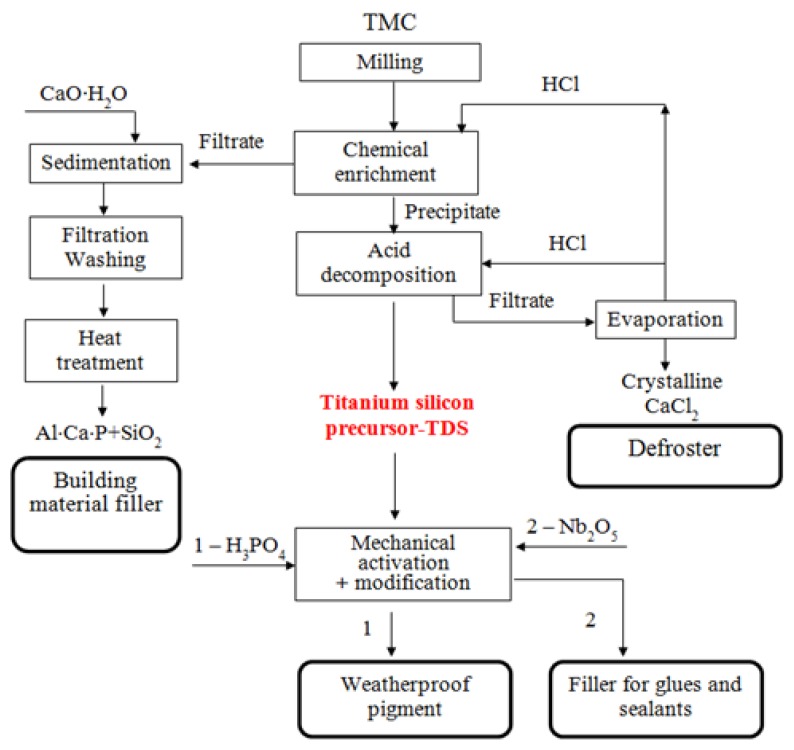
Technological scheme of processing titanium-containing mineral composition (TMC) to obtain weatherproof pigment and filler for glues and sealants.

**Figure 4 materials-13-01599-f004:**
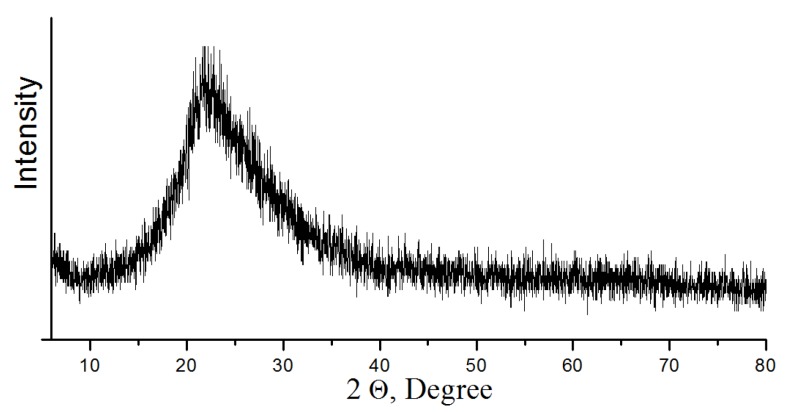
Powder X-ray diffraction pattern of composition (heat treatment at 500 °C).

**Figure 5 materials-13-01599-f005:**
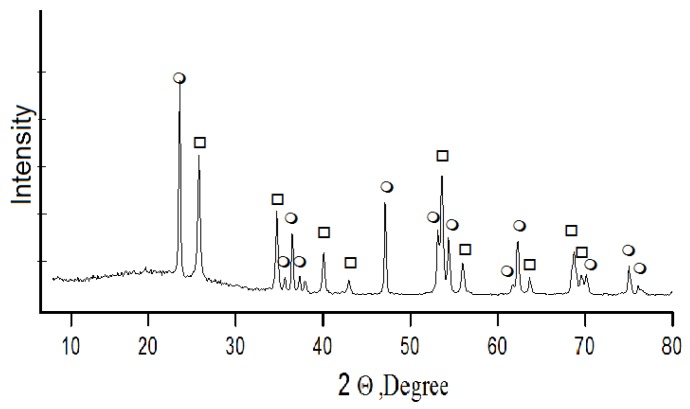
Powder X-ray diffraction pattern of DTK titanium dioxide in the form of ο—anatase, □—rutile.

**Figure 6 materials-13-01599-f006:**
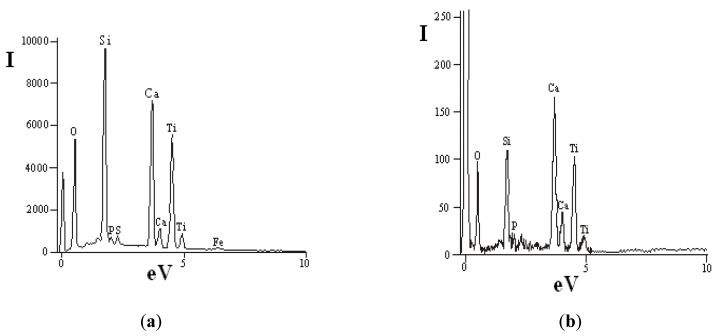
Energy-dispersive analysis of concentrate particle surface: (**a**) planetary mill; (**b**) initial concentrate.

**Figure 7 materials-13-01599-f007:**
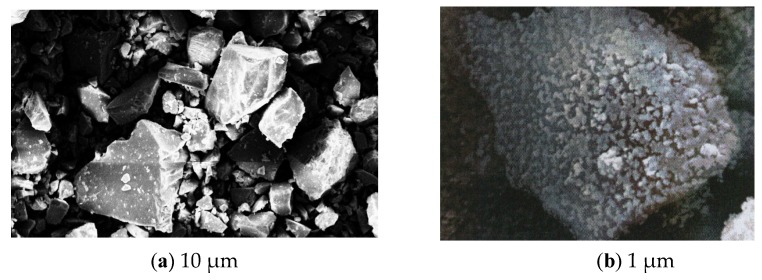
SEM images of titanite particles: (**a**)—before grinding; (**b**)—after grinding

**Figure 8 materials-13-01599-f008:**
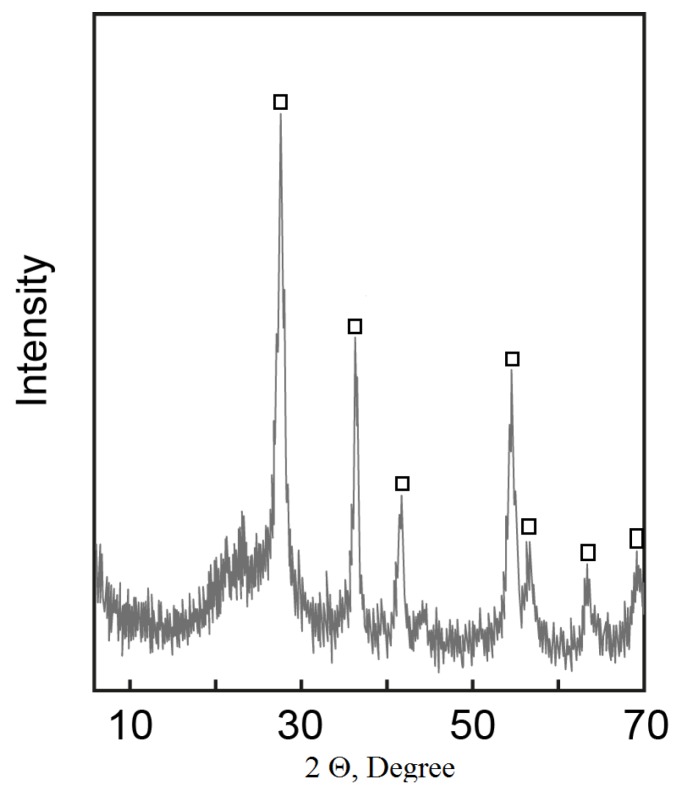
Powder X-Ray diffraction pattern of calcined material (TSK) □—rutile.

**Figure 9 materials-13-01599-f009:**
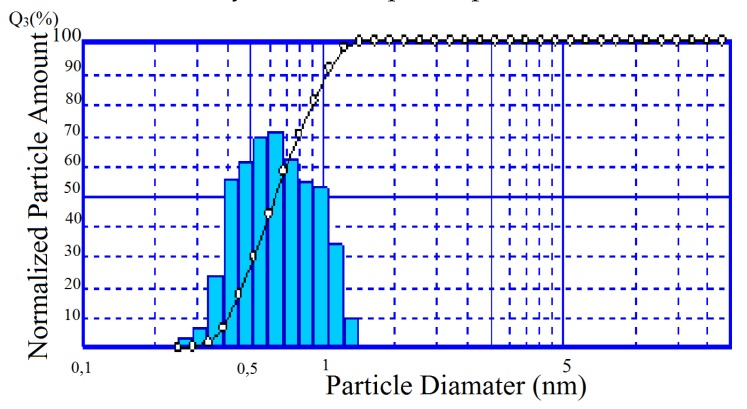
Particle size distribution in TSK.

**Table 1 materials-13-01599-t001:** The main minerals composing apatite-nepheline ores (ANO).

Mineral	Mineral Content in Ore, wt %	Mineral Formula
Apatite	33.7–35.0	Ca_5_(PO_4_)_3_F
Nepheline	40.6–42.2	(NaK)_2_OAI_2_O_3_·2SiO_2_
Aegirine	8.7–9.5	NaFeSiO_6_
Sphene	2.4–2.9	CaSiTiO_5_
Titano-magnetite	1.0–1.2	FeO·Fe_2_O_3_·TiO_2_

**Table 2 materials-13-01599-t002:** Conditions and results of the TMC chemical treatment (content of components at initial concentrate, wt %: TiO_2_—16.0; Al_2_O_3_—7.5; P_2_O_5_—6.0).

No.	Experimental Conditions	Content of Components at Titanite Concentrate, wt %		
		TiO_2_	Al_2_O_3_	P_2_O_5_
1	HCl—30 g/L, T:V_liq_ = 1:3, 2 h, t—20 °C	27.5	0.97	4.23
2	HCl—50 g/L, T:V_liq_ = 1:4, 2 h, t—20 °C	30.0	2.69	1.00
3	HCl—50 g/L, T:V_liq_ = 1:4, 2 h, t—40 °C	31.5	1.62	0.21

**Table 3 materials-13-01599-t003:** Surface properties of original and ultra-ground concentrate.

Surface Characteristics	Initial Titanite Concentrate	Titanite Concentrate after Grinding in Planetary Mill
S_spec._, m^2^/g	1.57	7.82
V_pore_, cm^3^/g	0.005	0.01
V_micropore_, cm^3^/g	0.00012	0.00013
S_micropore_, m^2^/g	0.30	0.33
R_av_, nm	25.68	15.71

**Table 4 materials-13-01599-t004:** TSK surface and technical properties.

Sample	S_spec._, m^2^/g	V, cm^3^/g	D, nm	pH	M, g/100 g	Refraction Index
TSK	10.05	0.068	16.48	6.8	28.7	1.85

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
