# Peer review of "Titanite-Containing Mineral Compositions and Their Chemical Treatment with Preparation of Functional Materials"

_materials, 2020, doi:10.3390/ma13071599_

Round 1

Reviewer 1 Report

1, The XRD characterization of raw materials should be added, this can further confirm the main composition and main phases of raw materials.

2, In the introduction, a statement of recovery methods and application of functional materials should also be added.

3, The contents in Figure is difficult to see, please revise the figures and format as required by the Journal.

Author Response

Response to Reviewer 1 Comments

Thanks to the reviewer for carefully reading the article. Our responses to reviewer comments.

Point 1. The XRD characterization of raw materials should be added, this can further confirm the main composition and main phases of raw materials.

Response 1.  According to the authors, it is not advisable to publish a XRD characterization of the initial mineral composition (TMC) in the text of the article. It is very difficult to identify the main and impurity mineral components in TMC due to its complex composition. In our opinion, the mineralogical composition given in the article contains enough information necessary for choosing the direction of research.

Point 2. In the introduction, a statement of recovery methods and application of functional materials should also be added.

Response 2. Information about the use of functional materials has been added in the introduction.

The utilization of the industrial wastes from the processing of multicomponent mineral ores is a more complicated and expensive task than storing them in the dumps. At the same time, maintenance of environmental standards for the functioning of enterprises is placed in the second place. Only in the Murmansk region, where the mining industry is developing rapidly, the amount of waste increases by millions of tons annually. The Khibiny deposits of complex apatite-nepheline ores (ANO) are the largest ones not only in Russia, but also in the whole world. Five main rock-forming minerals are specified in the ore: apatite, nepheline, sphene, aegirine and titano-magnetite (Table 1), which content varies within certain ranges depending on the ore-bearing rocks [1]. However, the modern ANO processing aims at more deeper recovery of apatite and partially nepheline. The rest minerals are almost not recovered due to the absence of demand for them; and therefore, they are disposed at tailings storage facilities [2].

Among the disposed minerals, calcium titanite-titanosilicate (СаTiSiO5) containing titanium represents a practical interest. Titanium metal and its compounds are diversely used in the industries producing products of various purpose.

Scientific and practical interest is associated with studies on titanite processing with preparation of various materials: non-toxic hardening agents [3, 4], mineral-like alkaline titanosilicates with skeleton structure [5-12] having sorption properties, functional fillers of plastic materials, rubber [13], sealant [14]. Mineral and synthetic waste products are widely used in the building industry as concrete aggregates [15], paints on the organic and water bases [16], special additives, in particular for imparting fire-retardant and photocatalytic properties for products. The use of industrial waste as fillers is covered in great detail in the monograph [17].

Point 3. The contents in Figure is difficult to see, please revise the figures and format as required by the Journal.

Response 3. The quality of Figures improved in accordance with the requirements of the Journal.

Kind regards,  article authors

Reviewer 2 Report

The manuscript about titanite-containing mineral compositions a waste from apatite-nepheline ore processing is clearly written, good organised, technology and findings are interesting. However, some remarks I have do make:

In general Figure captions, numbering and references in the text should be carefully checked

Line 15: diluted acid which one?

Line 22-21: I don’t understand the sentence especial: with obtained of modified materials

Figure 1: Low quality, markings are hardly readable

Line 51: why do you change the order?  41 – titanite to apatite – 20

Line 54-56: based on published data, which one? References should be given! Additional for the hardness I found 5-5,5 instead of your 6-6,5 please check this

Line 60-63: these lines would better fit after line 38

Figure 2: low quality, not really good to distinguish

Figure 3: Please check accuracy, schema shows more than the description in the text below. For example in the Figure you have 2 Filtrates (left and right), beneath you describe one Filtrate. Or after evaporation the HCl is reused for chemical enrichment (on the left side), is this true?

Line 86: for a better overview how many experiments were conducted, you should also refer to Table 2 at this point

Line 95: according to chemical data (Figure 4). If you refer to Figure 4 it is not chemical but mineralogical data! The next sentence “The use of the composition in the composition…” I don’t understand why you put this here in Material and Methods? However, delete it or provide more information’s, how concrete strength is increased and this should be experimental documented!

Figure 4: XRD composition is wrong used, better XRD pattern of total composition. Where does the 500 °C come from? It’s not explained in the text.

Line 108: Check Figure references in the text and Figure numbering: Figure 5 shows not a X-ray fluorescence analysis but a XRD pattern

Table 3: abbreviations e.g., Sspec should be explained below the table or in the text

Line 163-166: Sentence is hard to understand, due to quite long sentence and comma settlement

Line 177: VD-VA, VD-AK should be explained

Line 185-187: I am not quite sure what you wanted to say with these last two sentences.

Figure 8: the marking (unfilled square) should be explained and all obvious peaks should be marked

Line 204-207: index close to titanium dioxide a bit more details like references and number should be given

Author Response

Response to Reviewer 2 Comments

Thanks to the reviewer for carefully reading the article. Our responses to reviewer comments.

Point 1. Line 15: diluted acid which one?

Response 1.  diluted hydrochloric acid of couse

Point 2. Line 22-21: I don’t understand the sentence especial: with obtained of modified materials

Response 2.  We changed the sentence: Thermodynamic stability of activated particles is achieved by chemisorption or roasting.

Point 3. Figure 1: Low quality, markings are hardly readable

Response 3.  Figure 1 has been changed

Point 4. Line 51: why do you change the order? 41 – titanite to apatite – 20

Response 4.  It was misprints. We corrected them. 20 − apatite, 12 − aegirine

Point 5. Line 54-56: based on published data, which one? References should be given! Additional for the hardness I found 5-5,5 instead of your 6-6,5 please check this

Response 5.   We have given the references on published data about the main peaks on X-ray patterns of titanite samples [24].  Shchukina E.S., Gerasimov L.G., Ohrimenko R.F. Changing the composition of the titanium-containing polymineral mixture at its acid processing. Bulletin of the Murmansk State Technical University. 2013. V.16. â„–1. pp. 179-182. (in Russian)  Hardness by Moose 5-5.5 . Density- 3.4-3.54 g/sm3.

Point 6. Line 60-63: these lines would better fit after line 38

Response 6.  This paragraph has been moved to the introduction

Point 7. Figure 2: low quality, not really good to distinguish

Response 7. The quality of the Figure 2 is improved.

Point 8. Figure 3: Please check accuracy, schema shows more than the description in the text below. For example in the Figure you have 2 Filtrates (left and right), beneath you describe one Filtrate. Or after evaporation the HCl is reused for chemical enrichment (on the left side), is this true?

Response 8. The technological scheme is painted in the text of the article enough. During the acid decomposition of titanite concentrate, hydrochloric acid is intensively evaporated. The acid is trapped using a reflux condenser and can be reused in the chemical enrichment of the concentrate.

Point 9. Line 86: for a better overview how many experiments were conducted, you should also refer to Table 2 at this point

Response 9.  We have referred to Table 2

Point 10. Line 95: according to chemical data (Figure 4). If you refer to Figure 4 it is not chemical but mineralogical data! The next sentence “The use of the composition in the composition…” I don’t understand why you put this here in Material and Methods? However, delete it or provide more information’s, how concrete strength is increased and this should be experimental documented!

Response 10. We have given a more precise definition to text.

The multicomponent filtrate was disposed of to obtain an amorphous precipitate, which was calcined at 500°C. According to powder X-Ray diffraction analysis, the resulting composition consists of aluminum and calcium phosphates, as well as silica. The use of the composition in mixture of concrete increases its strength characteristics, which is confirmed by the authors of the publication.

Point 11.  Figure 4: XRD composition is wrong used, better XRD pattern of total composition. Where does the 500 °C come from? It’s not explained in the text.

Response 11.  We have written about it above and made the reference to the publication.

Point 12. Line 108: Check Figure references in the text and Figure numbering: Figure 5 shows not a X-ray fluorescence analysis but a XRD pattern

Response 12.  We have done it

Point 13. Table 3: abbreviations e.g., Sspec should be explained below the table or in the text

Response 13. The abbreviation has explained in the text below before the Table 4.

Point 14. Line 163-166: Sentence is hard to understand, due to quite long sentence and comma settlement

Response 14. We changed the sentence. Moreover, the specialized publications [28] note that this process is one of the most important in the pigment production technology. By selecting modifiers and conditions for their application, the properties of the final products can be varied over a wide range and their fields of application can be expanded accordingly.

Point 15 Line 177: VD-VA, VD-AK should be explained.

Response 15. These are brand of water-dispersion paints: WD-VA water-dispersion vinylacetate paint, WD-AC water-dispersion acrylic paint

Point 16 Line 185-187: I am not quite sure what you wanted to say with these last two sentences.

Response 16. We are talking about the synergistic effect of the additives

Point 17 Figure 8: the marking (unfilled square) should be explained and all obvious peaks should be marked

Response 17. We made changes

Point 18 Line 204-207: index close to titanium dioxide  a bit more details like references and number should be given

Response 18 We referred to the Technical terms of the titanium dioxide

Kind regards,  article authors.

Reviewer 3 Report

It is an interesting and original paper. The manuscript is generally clearly written and the paper deserves publication in the Materials journal. However, it must be improved by considering the following points:

(1) The paper deals the use of waste for the production of functional materials. Currently, wastes from various industries are used on a large scale. For this reason, the Introduction should discuss research with various types of useful wastes, which are presented in the literature.

Therefore the Introduction should be modified and a new section or paragraph about the waste materials used in construction (e.g.: additives, admixtures, recycled aggregates, glass, etc.) should be included in the manuscript. It is suggested that in the Introduction section these new papers should be discussed and cited:

Construction and Building Materials 213 (2019) 142–155, 

Construction and Building Materials 238 (2020) 117794,

Automation in Construction 112 (2020) 103111.

(2) The markings in figure 1 are out of focus.

(3) Colors from figure 3, i.e. yellow and green should be changed or removed. At present these colors make reading descriptions difficult.

(4) Please provide the number of counts in the figures: 5,6,8.

(5) The article lacks a description of SEM studies. Please provide more details about the microscopic examinations, eg: how large were the samples, how were the samples collected and prepared, on which microscope and at what magnifications the tests were carried out, what kind of the vacuum was used, etc.

(6) Figure 9 is illegible.

(7) Conclussions section is missing in the article. Please provide several conclusssions resulting from the conducted research.

Author Response

Response to Reviewer 3 Comments

Thanks to the reviewer for carefully reading the article. Our responses to reviewer comments.

Point 1. The paper deals the use of waste for the production of functional materials. Currently, wastes from various industries are used on a large scale. For this reason, the Introduction should discuss research with various types of useful wastes, which are presented in the literature.

Therefore the Introduction should be modified and a new section or paragraph about the waste materials used in construction (e.g.: additives, admixtures, recycled aggregates, glass, etc.) should be included in the manuscript. It is suggested that in the Introduction section these new papers should be discussed and cited.

Response 1. Information about the use of functional materials has been added in the introduction.

The utilization of the industrial wastes from the processing of multicomponent mineral ores is a more complicated and expensive task than storing them in the dumps. At the same time, maintenance of environmental standards for the functioning of enterprises is placed in the second place. Only in the Murmansk region, where the mining industry is developing rapidly, the amount of waste increases by millions of tons annually. The Khibiny deposits of complex apatite-nepheline ores (ANO) are the largest ones not only in Russia, but also in the whole world. Five main rock-forming minerals are specified in the ore: apatite, nepheline, sphene, aegirine and titano-magnetite (Table 1), which content varies within certain ranges depending on the ore-bearing rocks [1]. However, the modern ANO processing aims at more deeper recovery of apatite and partially nepheline. The rest minerals are almost not recovered due to the absence of demand for them; and therefore, they are disposed at tailings storage facilities [2].

Among the disposed minerals, calcium titanite-titanosilicate (СаTiSiO5) containing titanium represents a practical interest. Titanium metal and its compounds are diversely used in the industries producing products of various purpose.

Scientific and practical interest is associated with studies on titanite processing with preparation of various materials: non-toxic hardening agents [3, 4], mineral-like alkaline titanosilicates with skeleton structure [5-12] having sorption properties, functional fillers of plastic materials, rubber [13], sealant [14]. Mineral and synthetic waste products are widely used in the building industry as concrete aggregates [15], paints on the organic and water bases [16], special additives, in particular for imparting fire-retardant and photocatalytic properties for products. The use of industrial waste as fillers is covered in great detail in the monograph [17].

Point 2. The markings in figure 1 are out of focus.

Response 2. Figure 1 is changed

Point 3. Colors from figure 3, i.e. yellow and green should be changed or removed. At present these colors make reading descriptions difficult.

Response 3. Colors from Figure 3, i.e. yellow and green are removed

Point 4. The article lacks a description of SEM studies. Please provide more details about the microscopic examinations, eg: how large were the samples, how were the samples collected and prepared, on which microscope and at what magnifications the tests were carried out, what kind of the vacuum was used, etc.

Response 4.  This information you can read in the article line 144.

Point 5. Figure 9 is illegible.

Response 5.  Figure 9 is changed

Point 6. Conclusions section is missing in the article. Please provide several conclusssions resulting from the conducted research.

Response 6. We made the conclusion section in the article.

  • Almost all apatite deposits of the Khibiny massifs contain sufficient reserves of (apatite)-nepheline-titanite ore, which forms lens-like bodies up to 50 m thick and above 5 km long. This is a good titanium source that can be tapped without the traditional flotation schema, but by using only acidic cleaning from soluble impurities of apatite and nepheline;
  • A new technology of hydrochloric acidic processing of titanite is now developed. It allows extraction about 90 wt% of Ti and Si into hydrated titanosilicate precipitate TDS, while Ca remain in the chloride solution;
  • The calcium passing into the solution after acid decomposition of the titanite-containing concentrate was evaporated to obtain crystalline calcium chloride, which can be used as defroster. The hydrochloric acid is collected in the acid decomposition process, and can be used in circulation, thus the scheme becomes practically waste free;
  • The obtained hydrated titanosilicate precipitate is composed by titanium and silica oxides was used as precursor for preparation of a filler for glues and sealants of specialized application;

The finely-dispersed mineral pigment has been produced by the mechanical activation of the titanite-containing concentrate. The pigment was modified with titanium and calcium phosphates for improving its whiteness, water-resistance and weathering stability

Kind regards, autors article

Round 2

Reviewer 3 Report

I have no comments.